# Characterization of the Chloroplast Genome of *Argyranthemum frutescens* and a Comparison with Other Species in Anthemideae

**DOI:** 10.3390/genes13101720

**Published:** 2022-09-25

**Authors:** Yiran Zhao, Danyue Qu, Yueping Ma

**Affiliations:** College of Life and Health Sciences, Northeastern University, Shenyang 110004, China

**Keywords:** *Argyranthemum frutescens*, Anthemideae, chloroplast genome, ornamental plant, phylogenetic relationship

## Abstract

*Argyranthemum frutescens*, which belongs to the Anthemideae (Asteraceae), is widely cultivated as an ornamental plant. In this study, the complete chloroplast genome of *A. frutescens* was obtained based on the sequences generated by Illumina HiSeq. The chloroplast genome of *A. frutescens* was 149,626 base pairs (bp) in length, containing a pair of inverted repeats (IR, 24,510 bp) regions separated by a small single-copy (SSC, 18,352 bp) sequence and a large single-copy (LSC, 82,254 bp) sequence. The genome contained 132 genes, consisting of 85 coding DNA sequences, 37 tRNA genes, and 8 rRNA genes, with nineteen genes duplicated in the IR region. A comparison chloroplast genome analysis among ten species from the tribe of Anthemideae revealed that the chloroplast genome size varied, but the genome structure, gene content, and oligonucleotide repeats were highly conserved. Highly divergent regions, e.g., *ycf1*, *trnK*-*psbK*, *petN*-*psbM* intronic, were detected. Phylogenetic analysis supported *Argyranthemum* as a separate genus. The findings of this study will be helpful in the exploration of the phylogenetic relationships of the tribe of Anthemideae and contribute to the breeding improvement of *A. frutescens*.

## 1. Introduction

The chloroplasts of plants are self-replicating organelle that consists of their own genome, which encodes proteins that are essential for photosynthesis and different molecular processes [1,2]. The chloroplast genomes exhibit a highly conserved organization with a typical quadripartite structure that includes a pair of inverted repeats (IRs), separated by a large single-copy (LSC) region and a small single-copy (SSC) region among seed plants [3,4,5]. Beyond the conservation in the structure of the chloroplast genome, the differences, e.g., gene loss, sequence rearrangements, indels, and the expansion/contraction of the IR are found among species, even in individuals [6], which is valuable for plant evolution and taxonomic analysis [7,8]. In addition, chloroplast genomes are haploid in most angiosperms, and uni-parental inheritance [9,10] has made it a valuable resource for plant evolutionary and phylogenetic studies [11,12]. The complete chloroplast genomes have been demonstrated to be efficient, robust, and authentic in species discrimination and phylogenetic relationship resolution among species [5,10,13]. With the decreasing costs of next-generation sequencing, chloroplast genomes have been widely used to resolve the phylogenetic relationships of various plant species, e.g., in *Amaranthus* L. [14], *chrysanthemums* [5,15], and *Vitis* L. [16]. Comparative analyses of complete chloroplast genome could help us deepen penetrating insight into the evolution of chloroplast genomes and detect valuable polymorphic loci for phylogenetic analysis in plant [17,18].

Anthemideae belongs to Asteraceae, which contains 109 genera, including 1740 species [15]. There are abundant germplasm resources with excellent stress resistance, which is important for plant breeding, e.g., the improvement of cultivated chrysanthemums [19]. Although some progress has been made in the study of the relationship and geographical origin of some genera of Anthemideae based on morphology, cytology and molecular biology [20,21,22,23], the disputes about the relationship between the species and generic still existed due to the low sequence divergence of DNA markers and representative taxa across the family.

*Argyranthemum* is the largest endemic genus of the Atlantic oceanic islands in Anthemideae (Compositae) [20,24]. Taxonomic treatments recognized *Argyranthemum* as congeneric with *Chrysanthemum* [25]. Bremer and Humphries [20] considered *Argyranthemum*, *Chrysanthemum*, *Heteranthemis*, and *Ismelia* as a monophyletic group and treated as the subtribe Chrysantheminae. Molecular phylogenetic analyses based on *ITS* and cpDNA data supported *Argyranthemum* as a separate genus [26,27]. Thus, the phylogenetic position of *Argyranthemum* in the tribe remains uncertain. To provide more useful genetic data for resolving the phylogenetic relationship of *Argyranthemum* in Anthemideae, the complete genome sequence is a reliable resource.

To date, the genomic information of *Argyranthemum* is still lacking, hindering our understanding of its evolutionary history. Herein, the complete chloroplast genome was obtained from *A. frutescens*, a globally popular cultivated plant [28], on an Illumina HiSeq platform, and a comparison with nine other species in the tribe of Anthemideae was performed. The aims of this study were to: (1) characterize the chloroplast genome of *A. frutescens* and (2) explore the variation in the chloroplast genome among species of Anthemideae and determine the phylogenetic relationships of *A. frutescens*. This study provides useful genetic information to clarify the phylogenetic relationships of *Argyranthemun* and pave the way for the breeding and improvement of *A. frutescens*.

## 2. Materials and Methods

### 2.1. Sampling and Sequencing

The sample of *A. frutescens* was gathered from Shennongjia in the Hubei Province, China (31.26 N, 110.16 E), and was cultured in the nursery garden of Northeastern University, China. The voucher specimen was kept in the Herbarium of Northwest A&F University with the voucher number Zhao-2017–2. A modified CTAB method was used to isolated total genomic DNA from fresh leaves [29]. A Nanodrop 2000 spectrophotometer (Thermo Fisher Scientific, Waltham, MA, USA) and 0.8% agarose gel electrophoresis were used to assess the quality and quantity of DNA. Paired-end libraries of 500 bp inserts was constructed using Illumina TruSeq Library Preparation Kit (Illumina, San Diego, CA, USA) according to the manufacturer’s instruction. The library was sequenced on an Illumina HiSeq 2500 platform in Novogene (Tiananjing, China), and paired-end reads of 2 × 150 base pairs (bp) were generated.

### 2.2. Chloroplast Genome Assembly and Annotations

The raw sequencing data was filtered and trimmed using Trimmomatic v 0.32 [30] under default settings to remove adaptors and low-quality data. The quality of the filtered data was checked with FastQC. De novo and reference-guided methods were integrated to assemble the chloroplast genome as described by Ma et al. [5]. and Valcárcel and Wen [31]. The clean reads were initially de novo assembled into contigs using Velvet v.1.2.10 [32] implemented in Geneious 10.1.3 [33] with auto strategy. Furthermore, the reads were mapped to the chloroplast genome of *Ismelia carinata* downloaded from GenBank (NC_040110) using reference-guided assembly with default settings in Geneious. Then contigs obtained by de novo assembly mapped to the consensus sequence were obtained using the reference genome to check the errors or ambiguities resulting from either assembly method. The final chloroplast genome sequence of *A. frutescens* was obtained from the mapped contigs and annotated using GeSeq [34]. The start and stop codons of protein-coding genes and intron/exon boundaries were manually adjusted by comparison with other chloroplast genomes of Asteraceae from Genbank (e.g: NC020320, NC007977) using Geneious v.10.1.3. The annotated circular map was generated with the Organellar Genome DRAW tool (OGDRAW) [35] with default parameters and edited manually. The obtained chloroplasts genomes sequence of *A. frutescens* was deposited in the NCBI database (GenBank accession number OK585057)

### 2.3. Codon Usage and Putative RNA Editing Site

The relative synonymous codon usage (RSCU) of protein-coding genes was determined by CodonW1.4.2 [36] under default settings. Predictive RNA editors for the plant chloroplast (PREP-cp) was applied to explore the editing sites of putative RNA [37].

### 2.4. Chloroplast Genome Comparison

The chloroplast genome of nine species representing Anthemideae, including *Ajania variifolia*, *Artemisia argyi*, *Leucanthemella linearis*, *Leucanthemum maximum*, *Chrysanthemum zawadskii*, *I. carinata*, *Crossostephium chinense isolate JPBB*, *Neopallasia pectinate*, and *Soliva sessilis,* were downloaded from Genbank to compare with the chloroplast genome of *A. frutescens* for a deeper understanding of the intergeneric evolutionary relationships in species of the tribe of Anthemideae. MAFFT v5 [38] implemented in Geneious with default parameters was applied for pairing the sequence alignment of the chloroplast genomes. The complete chloroplast genome difference between *A. frutescens* and other species in the tribe of Anthemideae was determined with mVISTA in the shuffle-LAGAN mode [39]. The expansion and contraction of IRs junctions were performed by IRSCOPE [40].

### 2.5. Repeat Sequences and SSR Analysis

REPuter v1 (Bielefeld, Germany) [41] (https://bibiserv.cebitec.uni-bielefeld.de/reputer, accessed on 20 August 2022) was performed to detect long repetitive sequences in the cpDNA, including forward (F), palindromic (P), reverse (R), and complementary (C) repeats. The minimal repeat size was set to 20 bp; the hamming distance was set to 3 bp, and the sequence identity was >90%. Simple sequence repeats (SSRs) within chloroplast genomes of ten species were detected by MIcroSAtellite identification tool (Misa) [42] with a threshold of nine for mononucleotide; five for dinucleotide; four for trinucleotide; and three for tetranucleotide, pentanucleotide, and other nucleotide repeats.

### 2.6. Phylogenetic Analysis

The whole chloroplast genome sequences of 19 species in the tribe of Anthemideae were selected for phylogenetic analysis. Multiple-sequences alignment was performed using Geneious 10.1.3. The best-fit model was determined by MEGA11 [43]. A maximum likelihood (ML) tree was constructed with MEGA11 software using the model of GTR + G and 1000 bootstrap replicates, and *Helianthus annuus* was selected as the outgroup. All indels was excluded for ML analysis.

## 3. Results

### 3.1. Chloroplast Genome Assembly and Features of A. frutescens

Approximately 10 GB of raw data (2 × 150 bp in length) was generated on the Illumina HiSeq 2500. The chloroplast genome of *A. frutescens* was obtained by the integrated methods of de novo and reference-guided. The complete chloroplast genome of *A. frutescens* was 149,626 bp in length, with a typical quadripartite structure of seed plant chloroplast genomes. It comprised two inverted repeat regions (IRa/b, 24,510 bp) separated by a large single-copy region (LSC, 82,254 bp) and a small single-copy region (SSC, 18,352 bp) (Figure 1). The overall G + C content of the whole chloroplast, LSC, SSC, and IR regions is 37.5%, 35.6%, 30.9%, and 43.1%, respectively. The chloroplast genome contained 132 unique genes, including 85 protein-coding DNA sequences, 37 tRNA genes, and 8 rRNA genes. Seventeen genes were duplicated in the IR regions (Table 1). Nine of the unique genes and six of the tRNA genes contained a single intron. The genes coding clpP and ycf3 had two introns (Table 2). The rps12 was a trans-spliced gene with the first exon located in the LSC region, and other two exons located in IRs.

### 3.2. Codon Usage and Putative RNA Editing Site

The complete chloroplast genome of *A. frutescens* comprised 25,751 codons among 85 protein-coding genes. Leucine was the richest amino acid with 10.76% (2770) occurrence, and cysteine was the least-common codon with 1.11% (286). Most of the codons ending with A or T were presented as RSCU > 1 (Appendix A), suggesting that A or T were the preferred codon ending base.

Sixty-two putative RNA editing sites were detected in 23 CDS (Appendix A). The majority of editing sites were predicted in the *ndhB* (10 sites) gene and preferred to produce leucine. In addition, the second base of the codon showed a higher probability of alteration. However, most of the synonymous codons were always altered at the third base as we know, which suggested that RNA editing avoided invalid editing to some extent (Appendix A).

### 3.3. Repeat Sequences and Microsatellite Analysis

Long repetitive sequences and the SSR locus of the chloroplast genomes of *A. frutescens* and nine other species in the tribe of Anthemideae were analyzed. A total of 250 repeats were detected among these ten species, and F and P repeats were more abundant than C and R repeats. The total number and proportion of repeat types in 10 species showed similarity pattern, suggesting a similar evolutionary history and close relationship among species in the tribe of Anthemideae (Figure 2A). The most prevalent repeat units were in 26–30 bp, followed by 41–45 bp, while the length in 51–59 bp occurred less frequently (Figure 2B). Of which, twenty forward repeats, twenty-six palindromic repeats, three reverse repeats, and one complementary repeat were found in the *A. frutescens* chloroplast genome that was more similar to *I. carinata.* In *A. frutescens*, the longest forward repeat was 60 bp in length and was located in the IR region. In the LSC, IR, and SSC regions, 26, 8, and 15 repeats were presented, respectively.

A total of 1050 SSR loci were found across the chloroplast genomes of 10 species, with the shortest loci of 9 bp. *L. maximum* contained the highest number of SSRs (139), followed by *L. linearis* (114), *C. chinense isolate JPBB* (108), *N. pectinata* (108), *A. frutescens* (107), *I. carinata* (106), *C. zawadskii* (103), *A. argyi* (98), *A. variifolia* (95), and *S. sessilis* (72). In the chloroplast genome of the ten kinds of Anthemideae plants, mononucleotides were the prevalent SSR loci, which varied from 9 to 14 repeat units in length. Meanwhile, the abundance of pentanucleotide was the lowest. Additionally, *A. frutescens* and *I. carinata* contained the same number of trinucleotide (5), tetranucleotide (13) repeats, and pentanucleotide (1) (Figure 3), indicating a more recent evolutionary relationship between the two species. In all ten species, the SSRs were mainly composed of A and T, and mononucleotides repetition was the most prevalent in each chloroplast genome.

### 3.4. Genome Comparison and Boundaries between SC and IR Regions

The length of the chloroplast genome in ten plants varied considerably and ranged from 149,626 bp in *A. frutescens* to 151,865 bp in *L. maximum*. The whole genome exhibited a high degree of synteny and the conservation of structure and gene order (Appendix A),which indicated that the chloroplast genomes in tribe of Anthemideae were conserved. The variation among the 10 chloroplast genomes according to the analysis of mVISTA revealed that genes located in coding regions were more conserved than those in intergenic regions. Eleven highly polymorphic regions were detected including *ndhF*, *ycf1*, *ccsA,* and *ycf2* genes and *trnK-psbK*, *petN-psbM*, *atpI-atpH*, *trnT-psbD*, *ycf3-rps4*, *psbE-petL*, and *trnL-ndhF* intergenic regions (Appendix A). These regions might be potential molecular markers for phylogenetic analyses of the tribe Anthemideae.

The boundaries of IRs and SCs among the ten species from the tribe of Anthemideae were compared and showed a high resemblance. The junction between the LSC and IRa occurred within pseudo *rps19* with most part of *rps19* gene located in the IRa region, as is typical within angiosperms. Compared to the other nine species, the rps19 in the IRa of *L. maximum* was slightly contrasted (by about 28 bp) (Figure 4). The junction between SSC and IRb occurred within the *ycf1* gene, which was found in most land plants. The length of ycf1 varied apparently in different species and crossed over SSC and IRb. The junction of the SSC and IRa occurred between *ndhF* and pseudo *ycf1*. The pseudo *ycf1* did not extend to SSC in all species except for *A. frutescens*, and the *ndhF* located in SSC was 25–76 bp away from the junction in all species except for *S. sessilis*.

### 3.5. Phylogenetic Analysis

The phylogenetic position of *A. frutescens* was determined based on the complete chloroplast genome sequences of 19 species within the Anthemideae tribe, with *H. annuus* as outgroup. No heteroplasmy was detected among the chloroplast genome sequences we compared. The ML tree showed that all of the species in the tribe of Anthemideae were grouped into a monophyletic group with high support value. *A. frutescens* was clustered with *I. carinata* into a subclade with 100% supported value and sister with *L. maximum,* which was consistent with the taxonomic status (Figure 5). All species from the same genus were grouped into the same clade, which indicates that the chloroplast genome has a good species discrimation potential.

## 4. Discussion

In the present study, the entire chloroplast genome of *A. frutescens* was sequenced and assembled, and the comparison of the chloroplast genome between *A. frutescens* and nine other species in the tribe of Anthemideae was performed. The chloroplast genome of *A. frutescens* exhibited a classfic quadripartite structure like other seed plants [44]. The GC content of *A. frutescens* varied across different regions, and IR had the highest GC content (43.1%), as previously reported [45,46,47,48], suggesting high GC content (43.04%) increasing IR region stability and maintaining its structure in the cp genome evolution.

The codon usage bias in the plant chloroplast genome has been widely used to explore the evolutionary feature of many plant species [49,50]. Synonym codons for specific amino acids of genes was preferentially selected instead of random;y distributed [51,52]. The amino acid usage-frequency analysis in this study revealed that leucine was the most abundant amino acid and that cysteine was the least-occurrent codon. Codons ended with A or T were preferred in *A. frutescens,* which was consistent with reported genomes [53,54,55,56]. The RNA editing sites generally resulted in amino acid changes, which were useful for understanding the function of the translated proteins [57]. Most editing changes in the *A. frutescens* cp genome were C-to-U (88.4%) events. The majority of editing sites were predicted in the *ndhB* (10 sites) gene and preferred to produce leucine, which was consistent with previous reports [57,58], suggesting Leu may have a crucial role in the *A. frutescens* cp genome.

The repeat sequences and SSRs, which were widely distributed in cp genomes related to the genome rearrangement and recombination, are informative source for developing markers in population genetics and evolutionary studies [59]. The total number and proportion of repeat types in 10 species showed a similarity pattern (Figure 2A), suggesting a similar evolutionary history and a close relationship among species in the tribe of Anthemideae. Most of the SSRs were mononucleotides in the ten chloroplast genomes of Anthemideae plants and were mainly composed of A and T, as reported in previous studies of other Asteraceae [22,60]. The polymorphic long-repeat sequences and SSRs detected in our study might provide useful information for pursuing the evolution of species in the tribe Anthemideae in the future.

The whole plastome of *A. frutescens* and nine other species in the tribe of Anthemideae exhibited highly collinearity and was conservative in structure and gene order, similar to the previously reported plastid genome of Blumea species of Asteroideae [17]. However, the length varied considerably among ten species (Figure 4). Previous studies suggested that the change in genome size is mainly due to the length change of IR and SSC regions [61,62]. However the cp genome sizes in *Chrysanthemum boreale* and the cotton genus were found to be related to the length variation of LSC regions [21,63], whereas the size of cp genome in duckweeds was due to IR regions [64]. Compared to the other eight species, *A. frutescens* and *I. carinata* have the shorter length among ten species due to the shorter length of LSC and IR region (Figure 5). These findings suggest that the main reason for the change in chloroplast genome size may depend on the species.

The variation among the 10 chloroplast genomes according to the analysis of mVISTA revealed that genes located in coding regions were more conserved than those in intergenic regions, which was similar with the results in other plants [65,66]. The coding genes including *ndhF*, *ycf1*, *ccsA,* and *ycf2* genes were found to be relatively divergent as previous studies [67]. The relatively high variation of ycf1 have been reported in the genus of chrysanthemums [5,15]. Seven highly polymorphic regions in intergenic regions were also detected. These regions might be promising genetic markers for phylogenetic analyses of the tribe Anthemideae.

The variable pattern of the contraction and expansion of the inverse region in the plastid genome of plants are related to the size change and the reset of the four regions (LSC/IRB/SSC/IRA) boundaries. Minor differences were found to be presented in the boundaries of IRs and SCs among ten species of Anthemideae, which is consistent with the report of a previous study [46]. The variation in four region boundaries of *A. frutescens* was similar to *I. carinata*, supporting a close relationship between these two species.

The phylogenetic relationship among different genera in the Anthemideae has been explored based on chloroplast single-sequence data [25,26,27]. However, the position of genus *Argyranthemum* is still unclear. Phylogenetic analysis showed that all of the species in the tribe of Anthemideae were grouped into a monophyletic clade with 100% bootstrap support. *A. frutescens* was sister with *I. carinata*, supporting a close relationship between *Argyranthemum* and *Ismelia*, which was consistent with previous studies based on morphological and molecular evidence [23,24]. However, *A. frutescens* was not closely related with *Chrysanthemum*, supported it as a separate genus from *Chrysanthemum*. The species of the genera *Opisthopappus* and *Chrysanthemum* were grouped into a clade with 100% support value, supporting their close relationship that was consistant the AFLP results [68]. The genera of *Opisthopappus* and *Tanacetum* have been classified into the same subtribe of Tanacetinae, according to the morphology of pollen [69]; however, in this phylogenetic tree, the species of genus *Opisthopappus* was not closely related to *Tanacetum*. A broader taxon sampling and integration with other molecular makers, e.g., nuclear genes, need to be used to explore the phylogenetic relationship of tribe Anthemideae.

## 5. Conclusions

The complete chloroplast genomes of *A. frutescens* displayed the typical quadripartite structure of a land plant. A comparison chloroplast genome analysis among ten species from the tribe of Anthemideae revealed that the organization and gene order are highly conserved. Some repeated sequences, SSR loci, and highly variable regions were detected, which will be a potential molecular marker for phylogenetic analyses of the tribe Anthemideae. Phylogenetic analysis supported *Argyranthemum* as a separate genus. The findings of this study improve our understanding of the internal structure of the chloroplast genome of the tribe Anthemideae and are valuable for the future breeding, environmental adaptation, and hybridization of *A. frutescens*.

## Figures and Tables

**Figure 1 genes-13-01720-f001:**
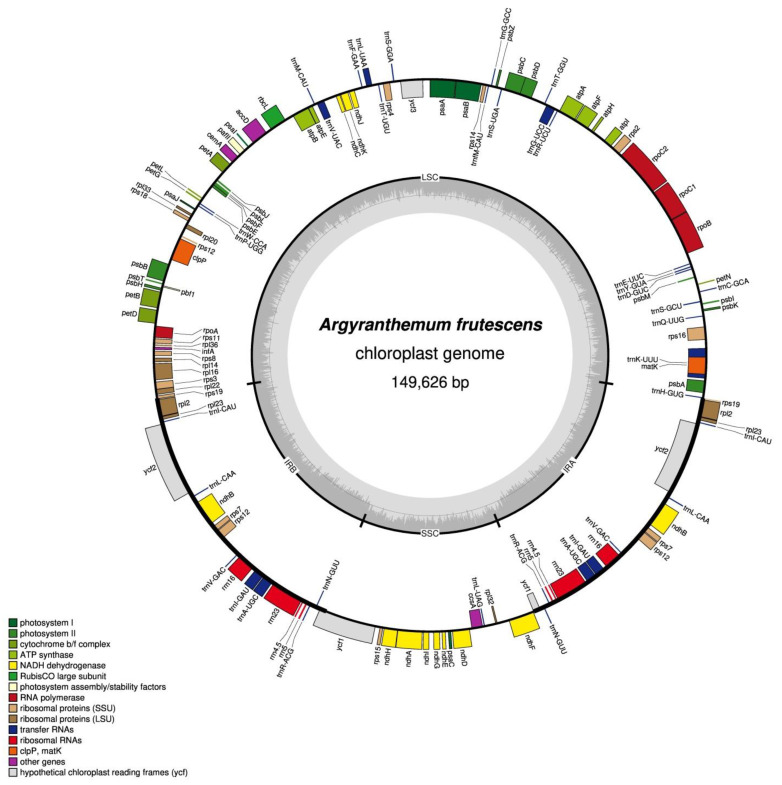
Circular chloroplast genome of the *A. frutescens*. Genes presented inside the circle were transcribed counter-clockwise, and those drawn outside were transcribed clockwise. Color-coding represents different gene functional groups.

**Figure 2 genes-13-01720-f002:**
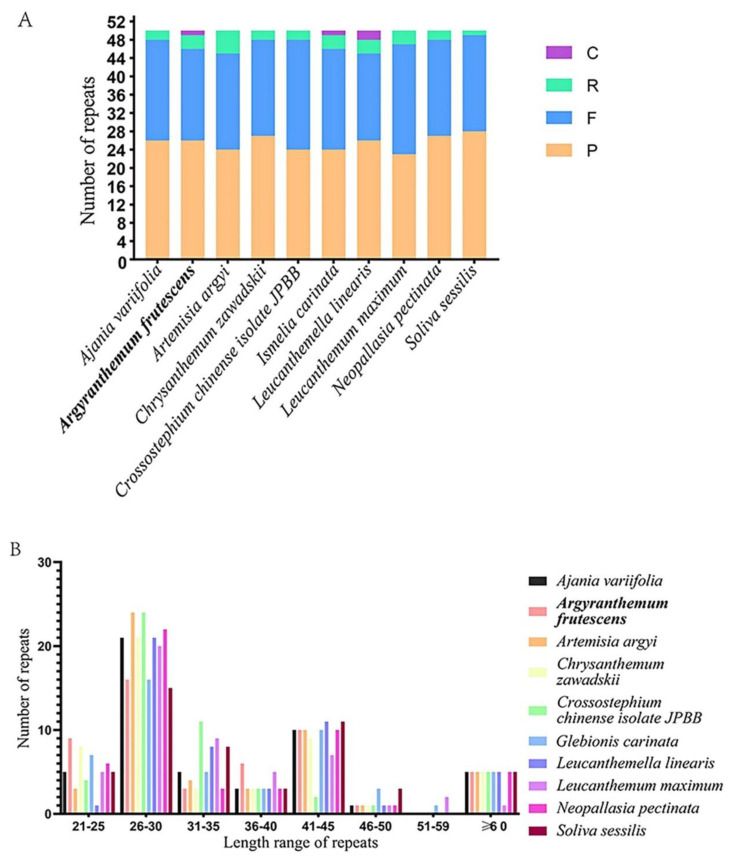
The long repetitive sequences of 10 chloroplast genomes. (**A**) Number of identified repeats. F, forward repeats; P, palindromic repeats; R, reverse repeats; C, complementary repeats. (**B**) Number of different repeat types of different lengths in 10 chloroplast genomes.

**Figure 3 genes-13-01720-f003:**
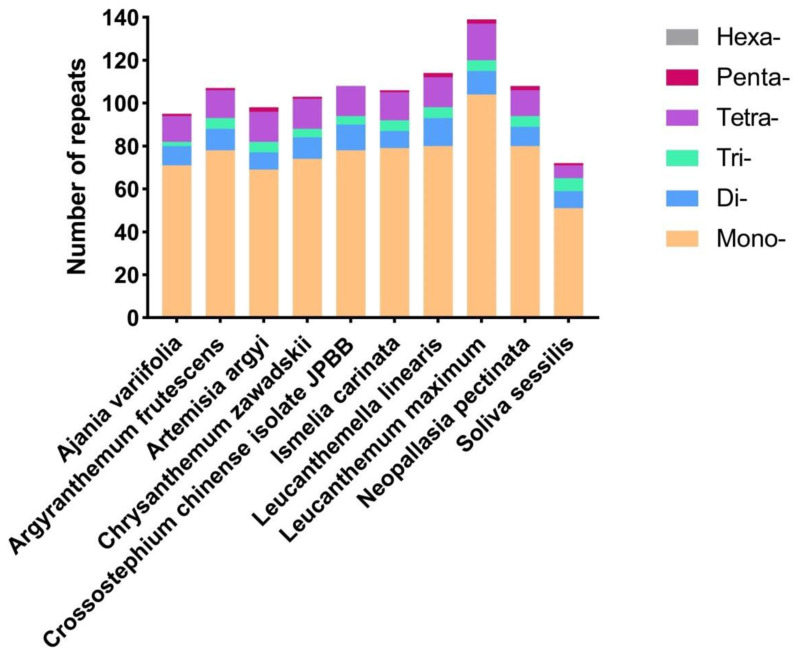
Simple-sequence repeats (SSRs) in ten different Anthemideae chloroplast genomes. Mono- represents mononucleotide SSRs, and Di- represents dinucleotide SSRs, etc. The vertical axis represents the number of different SSR types.

**Figure 4 genes-13-01720-f004:**
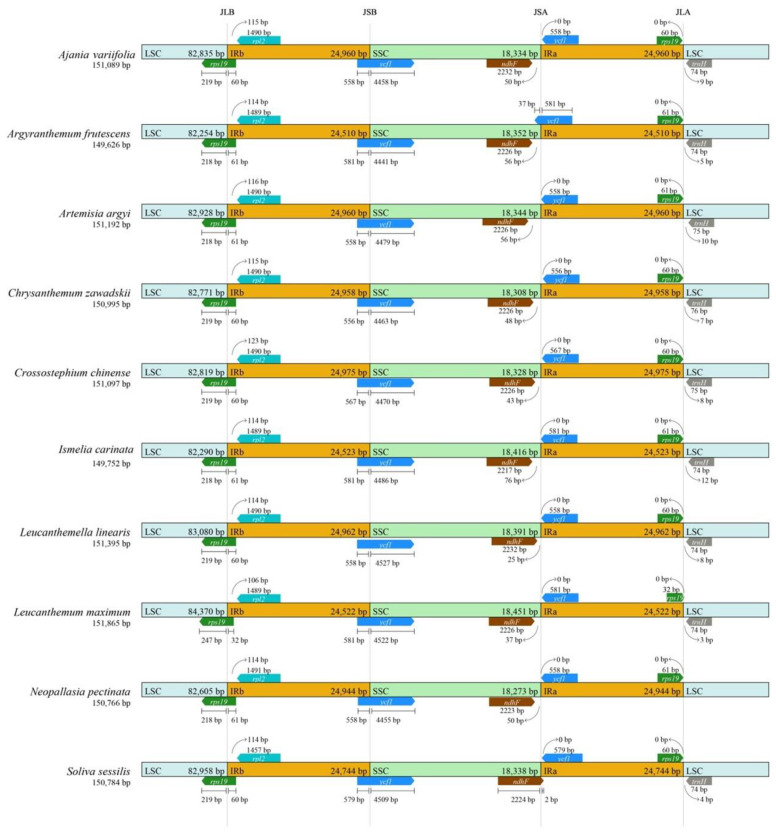
The junctions of IR/SC comparison among 10 species in the tribe of Anthemideae. Arrows indicated the distance of the gene to the junctions, and the I-shaped line represented the length of the gene on either side of the junctions.

**Figure 5 genes-13-01720-f005:**
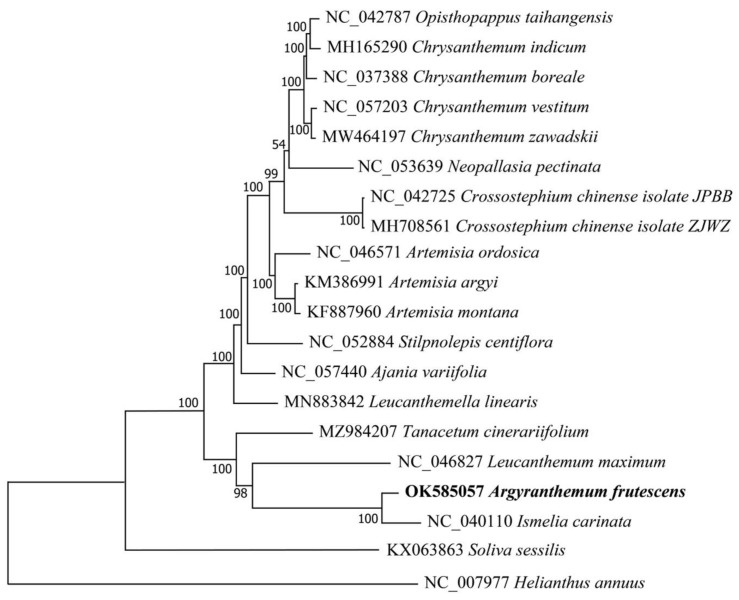
Maximum likelihood (ML) tree of 19 species in different genera of Anthemideae based on the whole chloroplast genomes.

**Table 1 genes-13-01720-t001:** Chloroplast genome general features of *A. frutescens*.

Characteristics		*A. frutescens*
**Size (base pair, bp)**		149,626
LSC length (bp)		82,254
SSC length (bp)		18,352
IR length (bp)		24,510
Number of genes		132
Number of protein-coding genes		85
Number of tRNA genes		37
Number of rRNA genes		8
Duplicate genes		17
G + C content	Total (%)	37.5
	LSC (%)	35.6
	SSC (%)	30.9
	IR (%)	43,1
	CDS (%)	37.8
	rRNA (%)	55.1
	tRNA (%)	52.8
	ALL gene %	39.4
Protein-coding part (CDS) (% bp)		51.6
All genes (% bp)		73.3
Non-coding region (% bp)		26.7

**Table 2 genes-13-01720-t002:** Genes annotated in the chloroplast genome of *A. frutescens*.

Category of Genes	Group of Genes	Gene Name
Photosynthesis-related genes	Photosystem II	*psbA, psbB, psbC, psbD, psbH, psbI, psbJ, psbK, psbL, psbM, psbT, psbZ*
	Large subunit of rubisco	*rbcL*
	Photosystem I	*psaA, psaB, psaC, psaI, psaJ,*
	NADH dehydrogenase	*ndhA *, ndhB ** (×2)*, ndhC, ndhD, ndhE, ndhF, ndhG, ndhH, ndhI, ndhJ, ndhK*
	Cytochrome c synthesis	*ccsA*
	Cytochrome b6/f complex	*petA, petB *, petD *, petG, petL, petN*
	cytochrome b559 subunit	*psbE psbF*
	Assembly/stability of photosystem I	*pafII*
	ATP synthase	*atpA, atpB, atpE, atpF *, atpH, atpI*
	Photosystem biogenesis	*Pbf1*
Transcription and translation related genes	Small subunit of ribosomal proteins	*rps11, rps12 ** (×2)*, rps14, rps15, rps16 *, rps18, rps19, rps2 *, rps3, rps4, rps7* (×2)*, rps8*
	Large subunit of ribosomal proteins	*rpl14, rpl16, rpl2* (×2)*, rpl20, rpl22, rpl23* (×2)*, rpl32, rpl33, rpl36*
	RNA polymerase subunits/transcription	*rpoA, rpoB, rpoC1 *, rpoC2*
	Translation initiation factor	*infA*
RNA genes	Ribosomal RNA	*rrn16* (×2)*, rrn23* (×2)*, rrn4.5* (×2)*, rrn5* (×2)
	Transfer RNA	*trnH-GUG, trnS-GCU, trnK-UUU *, trnV-GAC* (×2), *trnQ-UUG, trnC-GCA, trnD-GUC, trnR-ACG* (×2)*, trnY-GUA, trnE-UUC, trnR-UCU, trnA-UGC ** (×2)*, trnG-UCC *, trnT-GGU, trnS-UGA, trnI-CAU* (×2)*, trnG-GCC, trnM-CAU, trnS-GGA, trnL-CAA* (×2)*, trnT-UGU, trnL-UAA *, trnF-GAA, trnI-GAU ** (×2)*, trnV-UAC *, trnM-CAU, trnW-CCA, trnN-GUU* (×2)*, trnP-UGG, trnV-GAC, trnL-UAG*
Other genes	RNA processing	*matK*
	Hypothetical proteins	*ycf3 ***
	Product Clp (chloroplast)	*clpP ***
	Pseudo gene	*Ψycf1, Ψrps19*
	Acetyl-CoA carboxylase	*accD*
	Envelope membrane carbon uptake protein	*cemA*
Unknown function	Conserved open reading frames	*ycf1, ycf2* (×2)

Note: * Gene with one intron; ** Gene with two introns; (×2) Gene with two copies; Ψ indicates the pseudo gene.

## Data Availability

The data that support the findings of this study are openly available in the GenBank of NCBI at https://www.ncbi.nlm.nih.gov/, accessed on 20 August 2022), reference number (OK585057).

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
