# Peer review of "Characterization of the Chloroplast Genome of *Argyranthemum frutescens* and a Comparison with Other Species in Anthemideae"

_genes, 2022, doi:10.3390/genes13101720_

Round 1

Reviewer 1 Report

This is a good paper and the following revisions are required.

1. In some places, the English is incorrect. You need to check your English again.

Line 42, "a largest" should be "the largest".

Line 51, "o date" is incorrect.

There may be many other mistakes.

2. Line 70, Please note the name of the kit used to create the library.

3. Did you trim the results of the multiple alignment? If not, please perform it using trimal or equivalent.

4. Before calculating a phylogenetic tree using the maximum likelihood method, a model test is required. Please perform the model test with modeltest-ng or MEGA and create a phylogenetic tree based on the results.

5. After assembling the chloroplast genome, validate the assembly results by mapping short reads to the assembled sequence and show the results in a supplemental figure. Screen capture of the IGV is good as a figure. Were the mapping depths uniform? If not uniform, the assembly results need to be revalidated.

6. Was there heteroplasmy? A statement regarding the presence or absence of heteroplasmy is required in the text.

7. Please correct the line spacing (lines 127-135).

8. Figure 1 is small and difficult to view. Please make the figure as large as possible.

9. Figure 4 is small and difficult to view. Please make the figure as large as possible.

10. We also need a phylogenetic tree that shows branch lengths. Please change Figure 5 to a phylogenetic tree showing branch lengths or show a phylogenetic tree showing branch lengths as Figure 6.

Author Response

Dear reviewer,

Thank you for your valuable comments and suggestions. Accordingly, the manuscript has been rechecked and the necessary changes have been made. The responses to all the comments have been prepared and attached here. We look forward to working with you to move this manuscript closer to publication in Genes.

Thank you for your consideration. We look forward to hearing from you.

Sincerely,

Yueping Ma

Responds to reviewer comments.

  1. In some places, the English is incorrect. You need to check your English again. Line 42, "a largest" should be "the largest". Line 51, "o date" is incorrect. There may be many other mistakes.

Thank you for these comments. We checked our manuscript carefully and formatted it following the scientific ways

  1. Line 70, Please note the name of the kit used to create the library.

Thank you for this comment. We have revised it accordingly.

  1. Did you trim the results of the multiple alignment? If not, please perform it using trimal or equivalent.

 Thank you for this comment. Multiple sequences alignment was performed using Geneious 10.1.3. A maximum likelihood (ML) tree was constructed with MEGA11 software. The pairwise gaps were deleted. We have clarified it in the text, please see the text highlight in yellow.

  1. Before calculating a phylogenetic tree using the maximum likelihood method, a model test is required. Please perform the model test with modeltest-ng or MEGA and create a phylogenetic tree based on the results.

Thank you for this comment. The models applied was GTR + G, determined by MEGA 11, we have clarified it in the text. Please see the text highlight in yellow.

  1. After assembling the chloroplast genome, validate the assembly results by mapping short reads to the assembled sequence and show the results in a supplemental figure. Screen capture of the IGV is good as a figure. Were the mapping depths uniform? If not uniform, the assembly results need to be revalidated.

 Thank you for this comment. The figure of short reads mapping to the assembled sequence was shown in Figure S as a supplemental file, please see the figure S.

  1. Was there heteroplasmy? A statement regarding the presence or absence of heteroplasmy is required in the text.

Thank you for this comment. No heteroplasmy was detected among the chloroplast genome sequences we compared. We clarified it in the text, please see the text highlight in yellow.

  1. Please correct the line spacing (lines 127-135).

Thank you for this comment. We have revised it accordingly.

  1. Figure 1 is small and difficult to view. Please make the figure as large as possible.

Thank you for this comment. We have modified the size of Figure 1, please see the Figure.

  1. Figure 4 is small and difficult to view. Please make the figure as large as possible.

Thank you for this comment. We have modified the size of Figure 1, please see the Figure.

  1. We also need a phylogenetic tree that shows branch lengths. Please change Figure 5 to a phylogenetic tree showing branch lengths or show a phylogenetic tree showing branch lengths as Figure 6.

Thank you for this comment. The phylogenetic tree with branch lengths was provided in Figure 5.

Reviewer 2 Report

Zhao and Ma in this paper characterize the chloroplast genome of the Argyranthemum frutescens.  The abstract summarizes in a good way the content of the paper.

Introduction

The introduction sets nicely the questions they are trying to answer however it will be nice to include some more information as to why CP genomes or better say organelle genomes are advantageous in providing a nice phylogenetic picture of the Genus.

Another point is that the authors are missing a justification as to why Anthemidaeae is important to study.

In Sentence 50 they claim that the complete sequence is a reliable resource where do the authors base that assumption is there any relevant literature?

Materials and Methods

In paragraph 2.2  and through the rest of the paragraphs, the authors mention some software they have used for the assembly, they should mention whether they have used default parameters or if they have done anything special.

Results

The authors present the results in a nice way so that others can repeat their analysis. They only need to mention the parameters used. It will also be nice if authors can include a workflow of their analysis.

Discussion

In their discussion, the authors discuss their results in the context of the literature. However and since they compare their chloroplast genome with other genomes it will be nice to include some points on genome divergence , on hotspots if there any. What genes are the most variable and how this variability might be related with evolution.

Author Response

Dear reviewer,

Thank you for your valuable comments and suggestions. Accordingly, the manuscript has been rechecked and the necessary changes have been made. The responses to all the comments have been prepared and attached here. We look forward to working with you to move this manuscript closer to publication in Genes.

 Thank you for your consideration. We look forward to hearing from you.

Sincerely,

Yueping Ma

Responds to reviewer comments.

Introduction

The introduction sets nicely the questions they are trying to answer however it will be nice to include some more information as to why CP genomes or better say organelle genomes are advantageous in providing a nice phylogenetic picture of the Genus. Another point is that the authors are missing a justification as to why Anthemidaeae is important to study. In Sentence 50 they claim that the complete sequence is a reliable resource where do the authors base that assumption is there any relevant literature?

Thank you for this comment. Chloroplast genomes are known to be highly conserved in gene content, organization and structure among seed plant, are haploid in most flowering plants, and maternally inherited. The completed cp genome could provide more informative loci for phylogenetic analysis, which has been successfully used in species discrimination (Viljoen et al., 2018; Ma et al.,2020; Wen et al., 2018, 2020).We have added the advantageous and relevant literature using cp genome for phylogenetic analysis. Please see the text highlight in yellow.

Materials and Methods

In paragraph 2.2 and through the rest of the paragraphs, the authors mention some software they have used for the assembly, they should mention whether they have used default parameters or if they have done anything special.

Thank you for this comment. All the software were used under default settings, we have added the parameters of the software in the text of this part, please see the text highlight in yellow.

Results

The authors present the results in a nice way so that others can repeat their analysis. They only need to mention the parameters used. It will also be nice if authors can include a workflow of their analysis.

Thank you for this comment. The parameters used were all under defaut settings, we clarified it in the text of method, please see the text highlight in yellow.

Discussion

In their discussion, the authors discuss their results in the context of the literature. However and since they compare their chloroplast genome with other genomes it will be nice to include some points on genome divergence , on hotspots if there any. What genes are the most variable and how this variability might be related with evolution

Thank you for this comment. We have added some point about the divergence of genome, please see the text highlight in yellow.

Reviewer 3 Report

Dear author,  

    This study reported the chloroplast genome of A. frutescens, and generally describe the characteristic of cp genome. In the introduction, the relationship and the phylogenetic position of Argyranthemum in the tribe are still uncertain. However, the result about this content are limited, and similar with the result of previous report. The introduction and the result are not match with each other. The author should revised the introduction or the result. Moreover, the phylogenetic analysis was not presented in the abstract. Aside of those problems, the novelty of this ms was not enough, and the ms could provide limited information. The author should present more interesting result about the evolution.

Line 51: Spell mistake of “o date”

Lines 72-89: Previous study reported that the genes in chloroplast genome were from genome, pathogens, and mitochondrion (Maybe not accurate), many reads could both align to the genome and the reference-guided chloroplast genome. Using the method described in the ms, how to exclude the unfavorable factors causing by the characteristic of chloroplast genome?

The raw reads contained reads from chloroplast genome, and the sequencing depth of chloroplast genome was much high than that of the genome. Why not use this method to isolate the reads that belong to chloroplast genome, and de-no assembly the chloroplast genome?

Lines 96-101: Accession number should provide in the ms.

Line 192: ‘F and P repeats’ should not be abbreviated with first appearance in the ms.

Lines 234-243: What does this paragraph want to express? The boundaries of IRs and SCs among the ten species did not show obvious difference. If there is difference boundaries of IRs and SCs among the ten species, what the difference mean?

Author Response

Dear reviewer,

Thank you for your valuable comments and suggestions. Accordingly, the manuscript has been rechecked and the necessary changes have been made. The responses to all the comments have been prepared and attached here. We look forward to working with you to move this manuscript closer to publication in Genes.

Thank you for your consideration. We look forward to hearing from you.

Sincerely,

Yueping Ma

Responds to reviewer comments.

Comments and Suggestions for Authors

This study reported the chloroplast genome of A. frutescens, and generally describe the characteristic of cp genome. In the introduction, the relationship and the phylogenetic position of Argyranthemum in the tribe are still uncertain. However, the result about this content are limited, and similar with the result of previous report. The introduction and the result are not match with each other. The author should revised the introduction or the result. Moreover, the phylogenetic analysis was not presented in the abstract. Aside of those problems, the novelty of this ms was not enough, and the ms could provide limited information. The author should present more interesting result about the evolution.

Thank you for this comment. We have revised the introduction of our manuscript to make it consistent with the results.

Taxonomic treatments recognized Argyranthemum congeneric with Chrysanthemum, however in our results, Argyranthemum frutescens was not cluster with species of Chrysanthemum, supporting Argyranthemum as a separate genus from Chrysanthemum. Some detailed information were added in ms.

Line 51: Spell mistake of “o date”

Thank you for this comment. We are so sorry for this mistaken during the journal format transformation. The mistype has been revised.

Lines 72-89: Previous study reported that the genes in chloroplast genome were from genome, pathogens, and mitochondrion (Maybe not accurate), many reads could both align to the genome and the reference-guided chloroplast genome. Using the method described in the ms, how to exclude the unfavorable factors causing by the characteristic of chloroplast genome?

Thank you for this comment. This study wants to obtain the chloroplast genome sequence of A. frutescens. De novo and reference-guided methods were integrated to assemble the chloroplast genome. These method have been used to assemble chloroplast genome in many plant (Zhang et al.,2017; Valcárcel & Wen., 2019; Ma et al.,2020). The short reads were mapped to the assembled sequence to validate the assembly results. The mapped figure was provided as a supplementary figure, please see figure S.

Zhang, N., Erickson, D. L., Ramachandran, P., Ottesen, A. R., Timme, R. E., Funk, V. A., Luo, Y., Handy, S. M. (2017). An analysis of Echinacea chloroplast genomes: Implications for future botanical identification. Scientific reports, 7, 216.

Valcárcel, V., & Wen, J. (2019). Chloroplast phylogenomic data support Eocene amphi‐Pacific early radiation for the Asian Palmate core Araliaceae. Journal of Systematics and Evolution, 57, 547-560.

Ma, Y., Zhao, L., Zhang, W., Zhang, Y., Xing, X., Duan, X., Hu, J., Harris, A.J., Liu, P., Dai, S., Wen, J. (2020). Origins of cultivars of Chrysanthemum—Evidence from the chloroplast genome and nuclear LFY gene. Journal of Systematics and Evolution, 58, 925–944.

The raw reads contained reads from chloroplast genome, and the sequencing depth of chloroplast genome was much high than that of the genome. Why not use this method to isolate the reads that belong to chloroplast genome, and de-no assembly the chloroplast genome?

Thank you for this comment. I guess you want to know why we not assemble genome sequence. Direct sequencing of genomic DNA is still costly; in addition, the genetic background of A. frutescens is largely unknown now. However some works are on its way in our group, e.g. determine its genome size, flower development and so on. So getting chloroplast genome is one of these works. Of course assemble genome sequence is a good suggestion for our future work to explore.

Lines 96-101: Accession number should provide in the ms.

Thank you for this comment. The accession number was provided in the ms, please see the text in this part highlight in yellow.

Line 192: ‘F and P repeats’ should not be abbreviated with first appearance in the ms.

Thank you for this comment. The complete F and P were provided in the ms, please see the text highlight in yellow.

Lines 234-243: What does this paragraph want to express? The boundaries of IRs and SCs among the ten species did not show obvious difference. If there is difference of IRs and SCs among the ten species, what the difference mean?

Thank you for this comment. The variation of the size and rearrangement at the LSC/IRB/SSC/IRA boundaries in the chloroplast genomes can reflect the relationship between species to some extent, we have revised the text to make the sentence readily understandable. 

Round 2

Reviewer 2 Report

I have no further comments for the authos

Reviewer 3 Report

The revised ms has been improved in the introduction, all the question had been well response.